

# Drought intensity-duration-frequency curves based on deficit in precipitation and streamflow for water resources management

Yonca Cavus[1,2,3], Kerstin Stahl[3], Hafzullah Aksoy[4]

[1]Department of Civil Engineering, Beykent University, Istanbul, Turkey
[2]Graduate School, Istanbul Technical University, Istanbul, Turkey
[3]Faculty of Environment and Natural Resources, University of Freiburg, Freiburg, Germany
[4]Department of Civil Engineering, Istanbul Technical University, Istanbul, Turkey

*Correspondence to*: Yonca Cavus (yonca.cavus@hydrology.uni-freiburg.de)

**Abstract.** Drought estimates in terms of physically measurable variables such as precipitation deficit or streamflow deficit are key knowledge for an effective water management. How these deficits vary with the drought event severity indicated by commonly used standardized indices is often unclear. Drought characteristics assigned the same value in index are not necessarily the same in different regions, and in different months of the same region. We investigate drought to remove this disadvantage of the index-based drought IDF curves and develop intensity-duration-frequency (IDF) curves in terms of the

associated deficit. In order to study the variation of deficits, we use the link between precipitation and streamflow, and the associated indices, standardized precipitation index (SPI) and standardized streamflow index (SSI). More specifically, the analysis relies on frequency analysis combined with the total probability theorem applied to the critical drought severity. The critical drought has varying durations and it is extracted from dry periods. IDF curves in terms of precipitation and streamflow deficits for the most severe drought of each drought duration in each year are then subject to comparison of statistical

characteristics of droughts for different return periods. Precipitation and streamflow data from two catchments, the Seyhan River (Turkey) and the Kocher River (Germany) provide examples for two climatically and hydrologically different cases. A comparison of the two cases allows to test a similar method in different hydrological conditions. We found that precipitation and streamflow deficits vary systematically reflecting seasonality and the magnitude of precipitation and streamflow characteristics of the catchments. Deficits change from one month to another at a given station. Higher precipitation deficits

were observed in winter months compared to summer months. Additionally, we assessed observed past major droughts experienced in both catchments on the IDF curves which show that the major droughts have return periods at the order of 100 years at short durations. This coincides with the observation in the catchments and show the applicability of the IDF curves. The IDF curves can be considered a tool for using in a range of specific activities of agriculture, ecology, industry, energy, water supply etc. This is particularly important to end-users and decision-makers to act against the drought quickly and

precisely in a more physically understandable manner.

**Keywords.** drought intensity-duration-frequency curves, precipitation deficit, standardized precipitation index, standardized streamflow index, streamflow deficit



## 1 Introduction

Climate change increases extreme episodes, more severe and frequent droughts seem unavoidable and they are expected to

become unprecedented (Kreibich et al., 2022). Therefore, reliable drought analyses and estimations are needed to protect people's water demand in a sustainable way by mitigating the effect of water scarcity. Drought has been commonly assessed by using different types of standardized drought indices. These drought indices are derived from different variables each representing different types of droughts such as precipitation for meteorological drought, soil moisture for agricultural drought and streamflow for hydrological drought. Standardized drought indices are used by national meteorological and hydrological

services around the world due to their advantage of being non-dimensional variables, and they are useful for comparison of drought in different climate regions in terms of their characteristics such as duration, intensity, severity and return period. The separate use of the duration, severity/intensity and return period is not sufficient for a comprehensive drought characterization unless they are related in the form of severity/intensity-duration-frequency (S/IDF) curves. IDF curves reflect the statistical characteristics of variables and the relation among intensity and frequency for different durations at a station, providing rich

hydrological information in a single graph. Similar to precipitation IDF curves which are traditionally used in hydrological design, this study concentrates on drought IDF curves which can be proposed as useful tools in water management against low extremes.

Studies on the drought S/IDF curves are limited in the drought literature. In the existing literature, drought S/IDF curves have been developed mainly by using drought indices. As an early study, Dalezios et al. (2000) developed drought SDF curves

based on the Palmer drought severity index in the form of tables and isolines illustrating more severe droughts for longer return periods. Drought severity and duration were combined by a probabilistic model and copulas were applied on standardized precipitation index (SPI) to construct the joint distribution function and to establish the drought SDF curve in the form of return period isolines (Shiau and Modarres, 2009). In addition, performances of different copulas using various statistical methods were tested for the derivation of drought SDF curves based on SPI (Reddy and Ganguli, 2012). Todisco et al. (2013)

developed index-based SDF curves and integrated them with a methodology to account for the economic impact of drought. Halwatura et al. (2015) derived SDF curves at different time scales by using bivariate functions of duration and severity calculated from several drought indices. Most recently, Aksoy et al. (2021) focused on IDF curves of critical droughts based on SPI by using an empirical relationship between the drought intensity and its return period.

The drought curves described above are non-dimensional IDF or SDF curves based on standardized drought indices to assess

drought characteristics as in a few more studies (Gupta et al., 2020; Sahana et al., 2020; Pandya and Gontia, 2023). To date, only a few studies investigated drought S/IDF curves and their relation with physical drought indicators such as soil moisture, runoff, precipitation deficit and streamflow deficit, limiting the experience to a few specific regions. For example, Sung and Chung (2014) developed dimensional-drought SDF curves based on streamflow deficit using various threshold levels for water use in Korea. These drought S/IDF curves presented an insight into the use of deficits for a humid continental climate. Cavus





and Aksoy (2020) presented station-based S/IDF curves to quantify the recurrence intervals of critical droughts based on precipitation deficit for the Mediterranean climate.

We know that the same value of a drought index corresponds to different deficits in different regions. The non-dimensionality of drought indices comes at the expense of physical non-interpretability, i.e. most drought indices cannot be read quantitatively as actual precipitation and streamflow deficits. In climates with high seasonal variation (i.e., Mediterranean climate), the

difference between deficits varies greatly in each month while this difference may be lower in regions with low seasonality (i.e., humid climate). Determination of precipitation and streamflow deficits is a challenge when the common drought indices are used. Because meteorological and hydrological droughts correspond to temporal anomalies changing also spatially from one catchment to another and they are characterized based on long-term conditions, which are related to climatic and environmental factors (Vicente-Serrano et al., 2013; van Loon, 2015). Therefore, any non-dimensional drought severity or

intensity derived from index series might insufficiently represent the actual water availability for water management under drought conditions. In practice, deficits under drought conditions in different climates and catchments need to be quantified and be linked with the duration, deficit volume and return periods. With SDF or IDF curves we can construct this relation to quantify drought characteristics and obtain knowledge about past drought events for practical consideration, which is the motivation of this study to derive deficit IDF curves.

The overarching objective is to develop drought IDF curves based on precipitation and streamflow deficits at different time scales and appraise their usefulness in different climates. For this, we convert drought index-derived characteristics into precipitation and streamflow deficits for different cases. For comparison, we consider two catchments in different climatic regions. Hence, the comparison between the regions and among the variables we used provides the framework of analysis. Specifically, we aim to assess how the deficits vary in the considered study regions and finally explore how deficit IDF curves

might be used for different purposes. We address the following research questions:

1. How different are deficit IDF curves from their non-dimensional index-based counterparts?

2. What is the relation between drought indices and indicators? More specifically, how do deficits in precipitation and streamflow change with the associated standardized drought index in time and space?

3. For what purposes might deficit-IDF curves be used in water management practice?

**2 Case study catchments and Data**

Precipitation and streamflow data from meteorological and streamflow gauging stations of two regions, one in Turkey and another in Germany, were used. We selected Seyhan River Basin in southern Turkey and Kocher catchment in southwestern Germany. Seyhan River is important for the production of hydroelectric energy and the basin itself for irrigated agricultural activities, which increases water demand under low water availability (GDWM, 2019). The Kocher River is important for

agricultural activity and hydropower production from run-of-the-river power plants (Zeitler et al., 2017). Agricultural land covers about 51% of the catchment area and it has also settlement areas and forest (Bardossy, 2007).





The selected gauging stations from the headwaters of these basins are a subset of the benchmark catchments with good hydrometric performance and nearly natural flow conditions. The selected stations have long-record length dataset which is

crucial for correctly characterizing the drought and estimating the long return period-droughts (Table 1). The data of the selected catchments come from various sources. For Turkey monthly precipitation data of station-based observation come from the State Meteorological Service (MGM). Daily streamflow data were downloaded from the State Hydraulic Works (DSI) website. For Germany, daily precipitation data were downloaded from the climate data center of the German Weather Service (DWD) website. Streamflow data of this catchment were acquired from the Environment Agency of Baden-Württemberg

(LUBW).

The seasonality in precipitation and streamflow of Seyhan follows a typical Mediterranean climate with a wet winter and a dry summer. It produces seasonal precipitation and streamflow variation with peaks in December and April, respectively, and a low precipitation and flow period extending over the summer (Figure 1). The seasonality of precipitation in the Kocher catchment varies over the year with higher precipitation in winter and summer, less precipitation in spring and autumn.

Streamflow varies within the year with high flows in winter and low flows in summer. Flow duration curves (FDC) show the Kocher can sustain flow longer and then decreases sharply at the lower end, while the Seyhan gradually approaches the minimum. This is a distinct difference in the important lower part of the FDC's. In the long-term average daily streamflow in Seyhan, maximum values concentrated in winter. In Kocher, high flows are likely to observe throughout the year except for a period from July to September. However, in this period, maximum values are still higher than the long-term average of the

daily streamflow.

**Table 1.** Characteristics of selected meteorological and streamflow gauging stations in Seyhan and Kocher

| Meteorological Station | | | | | | | |
|---|---|---|---|---|---|---|---|
| Region | Code | Name | Latitude Longitude | Altitude (m above m.a.s.l) | Record period | No Precipitation Months (%) | Average Annual Precipitation (mm) |
| Seyhan | 17351 | Adana | 37.00 N 35.34 E | 23 | 1960-2016 | 11 | 662.74 |
| Kocher | 01674 | Göggingen | 48.86 N 9.89 E | 490 | 1952-2020 | 0 | 982.83 |
| Streamflow Gauging Station | | | | | | | |
| Region | Code | Name | Latitude Longitude | Altitude (m above m.a.s.l) | Record period | Drainage Area (km$^2$) | Average Specific Discharge (l/s-km$^2$) |
| Seyhan | E18A001 | Göksu Himmetli | 36.04 N 37.86 E | 665 | 1953-2011 | 2596.8 | 6.01 |
| Kocher | 3465 | Stein | 49.25 N 9.28 E | 189 | 1979-2021 | 1928.73 | 8.45 |



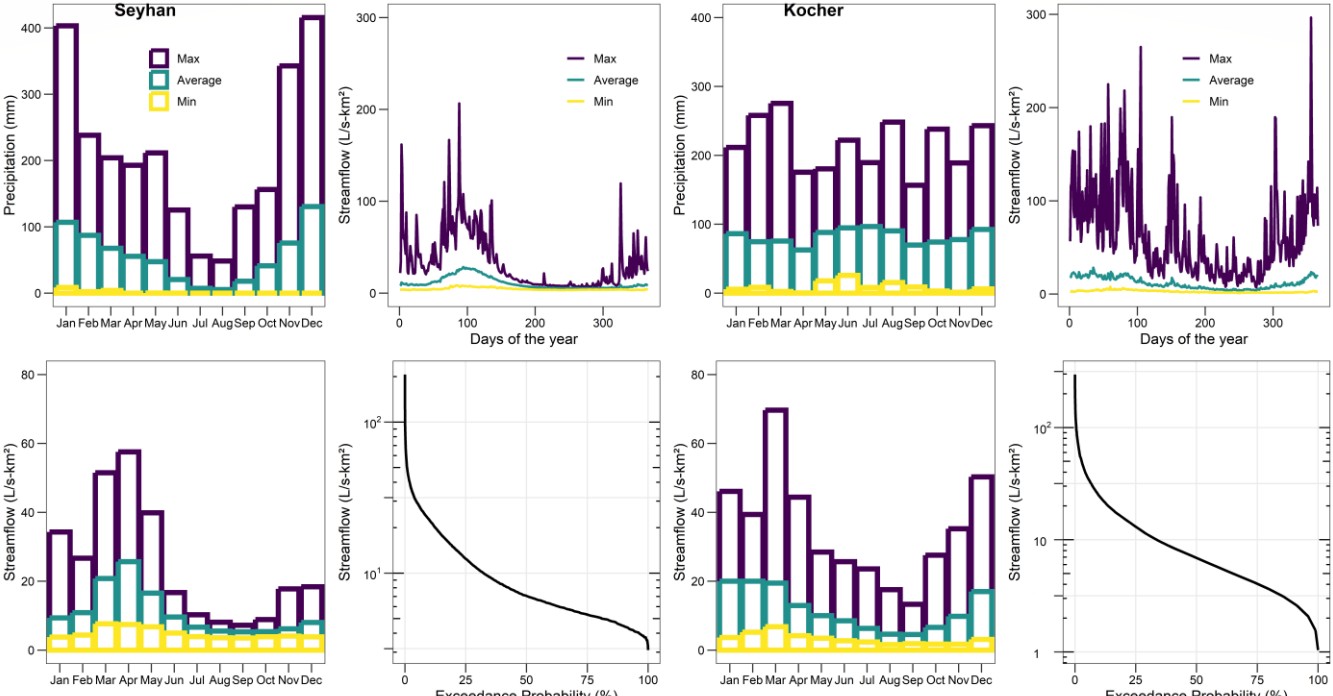

**Figure 1.** Seasonal characteristics of the meteorology and streamflow gauging stations with their flow duration curves for
Seyhan and Kocher

## 3 Method

This study follows several methodological steps (Figure 2): (1) Calculation of standardized drought indices, (2) Determination
of the most severe (critical) droughts for each drought duration, (3) Application of frequency analysis with the total probability
theorem to critical drought severity, (4) Calculation of precipitation and streamflow deficits by using logistic curve fitting, (5)
Derivation of drought IDF curves based on precipitation/streamflow deficits, (6) Comparative analysis of IDF curves.





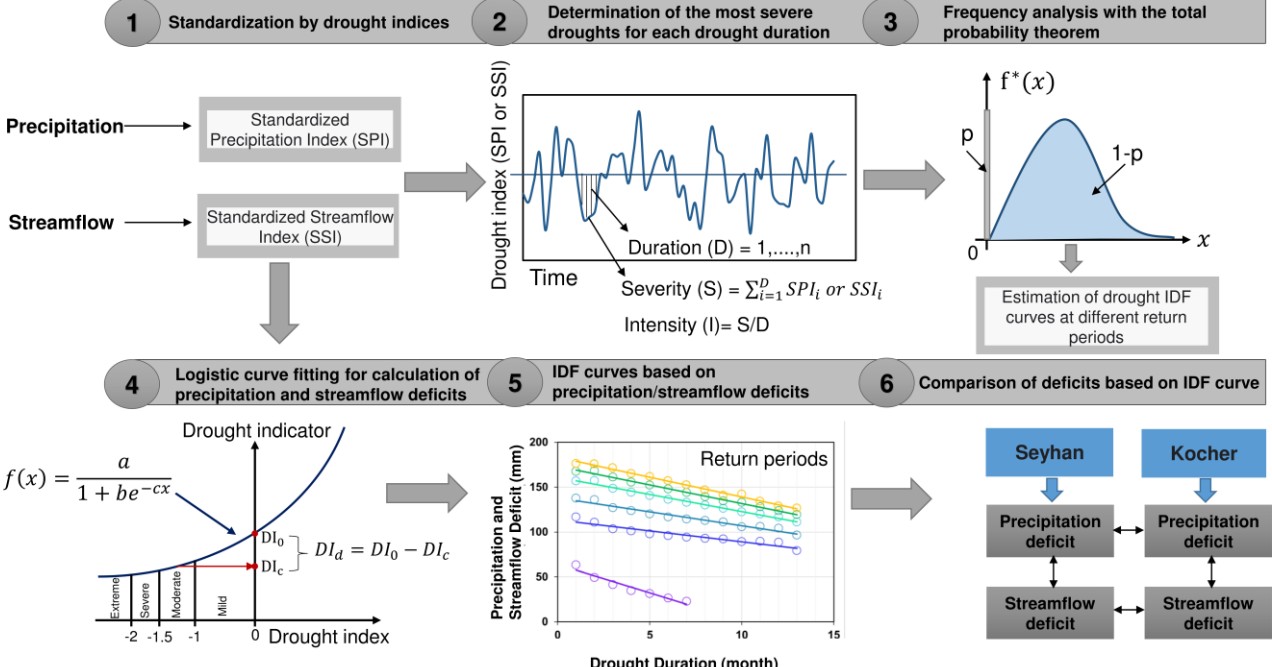

**Figure 2.** Illustration of the methodology used to develop drought IDF curves based on precipitation deficit and streamflow deficit

Step 1 involves standardization by drought indices. To characterize the drought, we used the Standardized Precipitation Index (SPI; McKee et al., 1993) and the Standardized Streamflow Index (SSI; Vicente-Serrano et al., 2012) (Figure 2). They are well-established drought indices which are widely used for the drought quantification. The SPI was calculated using monthly precipitation data accumulated at 1, 3, 6, 12-month time scales. Similarly, SSI time series of the same time scales were calculated to detect the total aggregated streamflow deficit. Precipitation was used as the monthly total precipitation for SPI

while streamflow was taken as the volume of monthly average streamflow for SSI. The Gamma probability distribution function is fitted to the precipitation data and transformed to the standard normal distribution with zero mean and unit standard deviation. We found the General Extreme Value (GEV) as the best-fit probability distribution function of the streamflow volume accumulated over a month for the two catchments. The length of the streamflow record and the selected probability distribution function strongly affect the calculated SSI (Wu et al., 2005, Tijdeman et al., 2020). To allow a comparison of the

streamflow deficits in the two different case study rivers in different climate regions, here we used the same distribution for both catchments.

In Step 2, dry periods were identified from the SPI and SSI time series (Figure 2). Any period of time with SPI and SSI values lower than zero (SPI < 0 or SSI < 0) was considered as a dry period. Based on the concepts of Cavus and Aksoy (2020), dry



periods consist of droughts; a dry period has a fixed length and a drought has a duration. Drought duration changes from 1

month to the dry period length. A drought with duration shorter than the dry period length is a within-period drought and a

drought with duration that cover the whole dry period length is a singular drought. The sum of consecutive negative values of

the drought index for each drought duration is the drought severity. The drought intensity is the drought severity divided by

the drought duration. In any year, drought of a given duration with maximum severity was taken as the critical drought for

each particular duration. When only one drought exists within one year, no matter how long and how severe it is, we assigned

it as the critical drought to this particular year. A year in which drought is not observed is called the no-drought year, and for

such a year, the critical drought severity is zero. For further definitions and explanations of these drought concepts see Cavus

and Aksoy (2019, 2020) and Aksoy et al. (2021).

In Step 3, frequency analysis was applied on the critical drought severity of each duration for each year at each time scale

(Figure 2). In no-drought years, the critical drought severity is zero as these years are totally covered by a wet period. Thus,

the critical drought severity time series has zero and non-zero values. Frequency analysis was applied on non-zero critical

drought severity time series 10-year long at minimum from which zero values were removed. Zero values in the drought

severity time series have a probability of occurrence while the non-zero values have a probability distribution function. The

total probability theorem described in the literature (Haan, 1977, Aksoy et al., 2021) was applied to data sets with zero and

non-zero values. The best-fit probability distribution function of the non-zero critical drought severity was determined by

taking into account the occurrence probability of zero values. We considered the 2- and 3- parameter Gamma (G2, G3), the

Generalized Extreme Value (GEV), the 2- and 3- parameter log-normal (LN2, LN3), Log-Pearson Type 3 (LP3), and the 2-

and 3- parameter Weibull (W2, W3) probability distribution functions which are widely used in literature. We used absolute

values of the critical drought severity in the frequency analysis as the probability distribution functions are only expressed for

positive variables. We decided on the best-fit probability distribution function after the Anderson-Darling statistical test. The

critical drought severities corresponding to 2, 5, 10, 25, 50 and 100-year return periods were calculated for all drought durations

by using the fitted probability distribution function of the relevant drought duration.

As SPI and SSI are calculated from precipitation and streamflow, respectively, a strong relationship can be expected between

the drought indicators (precipitation and streamflow) and the associated drought indices (SPI and SSI). This relationship is

determined in Step 4 by applying curve fitting in the form of nonlinear regression (Figure 2). To establish a functional

relationship between the drought indicators and the drought indices, we tested different curves among which the Logistic

curve, which is also known as sigmoid model, was selected as it provided the highest correlation (Sit and Poulin-Costello,

1994). It is given by

$$f(x) = \frac{a}{1+be^{-cx}} \tag{1}$$

in which $x$ is the drought index, and $a$, $b$ and $c$ are parameters.

In Step 5, drought indicators were converted to precipitation and streamflow deficits by using the relation between the drought

indicators and indices. The critical drought severity corresponding to each drought duration and return period were inserted





into the generalized logistic function (Equation 1) to find the critical precipitation or critical streamflow value (DIc). The precipitation or streamflow value corresponding to the maximum drought severity was calculated from the logistic function. We consider SPI = 0 and SSI = 0 as the threshold value of the drought index for all time scales. Threshold values of drought

indices were inserted into the logistic function to calculate the threshold drought indicator ($DI_0$). The difference between the threshold drought indicator and the critical drought severity of a given duration and return period is the precipitation deficit or streamflow deficit, which are calculated by

$$DI_d = DI_0 - DI_c \qquad (2)$$

Based on the calculated precipitation and streamflow deficits, drought IDF curves of the meteorological and streamflow

gauging stations were plotted at different time scales. These deficits provide knowledge to quantify drought IDF curves in terms of physical variables, which are the precipitation deficit and streamflow deficit.

Finally, in Step 6, drought IDF curves are compared between precipitation and streamflow deficits to determine how precipitation deficit propagates to streamflow deficit in these two catchments and how the deficits change from one catchment to another. Additionally, for the assessment of the applicability of the deficit-IDF curves we evaluated one specific drought

events observed in each catchment. A severe drought hit the Eastern Mediterranean in 2008, particularly its southern part where the Seyhan River Basin is located (GDWM, 2019; Cavus et al., 2022). The Kocher catchment was affected by the drought event of 2018 in Central Europe (Brunner et al., 2019; Tijdeman et al., 2022; Rakovec et al., 2022). To show the characteristics of these major droughts on the deficit-IDF curves, we identified months with the highest severity accumulated over 12 months for droughts of different durations from D = 1 month to the longest duration in the IDF curves. We considered

the 12-month time scale IDF curves for this demonstration. For each duration, the precipitation deficit was calculated by taking the difference between the 12-month precipitation accumulation and precipitation threshold of the precipitation record. The largest value of precipitation deficits calculated as the critical value of each duration was then replaced on the deficit IDF curves to derive the return period of the observed drought.

## 4 Results

### 4.1 Drought indices and indicators

The time series of SPI show a high variability with frequent fluctuations at shorter time scales. At longer time scales, the time series become smoother with less fluctuations (Figure 3). At the meteorological station in the Seyhan basin, dry periods become more visible with increasing time scale at the beginning of the time series. A similar observation can be made for the Kocher catchment. SPI time series at Kocher show lower negative values than those at Seyhan. Higher variability and more frequent

fluctuations are also evident in the SSI time series at short time scales while they become smoother with lower variability and less frequent extreme values at longer time scales (Figure 3). However, the SSI time series fluctuates less frequently compared to the SPI time series. Major dry periods in Seyhan are visible in 1974, 1990, 2008 and 2014 in the SPI and SSI time series,





particularly when the time scale increases. Drought events in the Kocher catchment are visible in 1990 and 2018 in the SPI and SSI time series.

The previously described seasonality in general precipitation (Section 2) is also evident in the relation between precipitation and the drought index, SPI, at 1-, 3- and 6-month time scales for which 12 curves were obtained (Figure 4). At the annual time scale, the relation between precipitation and SPI is represented by one a single curve as the annual precipitation has no seasonality. In comparison, the precipitation seasonality in Kocher is not as dominant as that in Seyhan and it becomes less pronounced with increase in the time scale. Therefore, at the 1-, 3-, and 6-month time scales, monthly curves between

precipitation and SPI are less variable due to less seasonal precipitation in Kocher than Seyhan. Because of the climate characteristics (given by the base climatology), the derived drought characteristics differ. Similar to the case for precipitation, Figure 4 clearly shows that the relation between streamflow and SSI changes from month to month at time scales of 1, 3 and 6 months due to the within-year seasonality of streamflow. Therefore, separate equations of each month should be used at sub-annual time scales. One equation can be used for the 12-month time scale as the seasonality in the streamflow is removed at

the annual scale. The variability in the curves that were fitted to obtain the relation between streamflow and SSI shows the seasonality in the streamflow of Kocher catchment although it is still less compared to the variability in the curves of Seyhan.



**Figure 3.** The SPI and SSI time series at 1-, 3-, 6- and 12-month time scales for Seyhan and Kocher





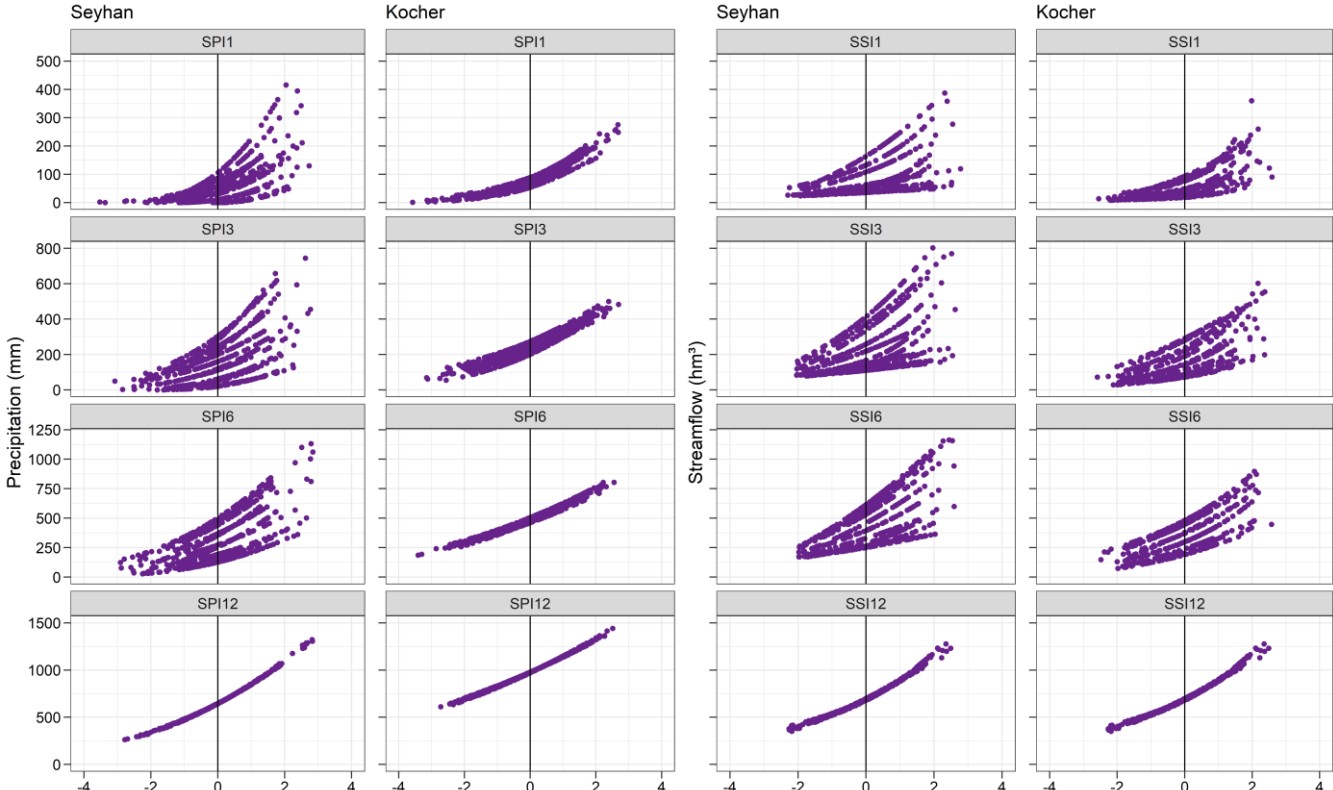

**Figure 4.** The relation between precipitation as drought indicator and SPI, and precipitation as drought indicator and SSI at 1-
, 3-, 6- and 12-month time scales for Seyhan and Kocher

**4.2 Frequency analysis of critical drought severity**

Frequency analysis of the critical drought severity shows that GEV is the best-fit probability distribution function for the
majority of cases when all droughts at all the time scales are considered in the meteorological and hydrological data records
(Table 2). The LP3, W2, LN2 and G2 distributions can be considered as alternative distributions for a few cases among which
LP3 is the second-best distribution for the critical drought severities calculated from SPI12 in both catchments. At this
particular time scale of SPI, the best-fit probability distribution function tends to change from GEV to an alternative probability
distribution function for long-drought durations.





**Table 2.** The best-fit probability distribution functions of the critical drought intensities of different durations for the time scales used

| Drought Duration (month) | Seyhan | | | | Kocher | | | | Seyhan | | | | Kocher | | | |
|---|---|---|---|---|---|---|---|---|---|---|---|---|---|---|---|---|
| | SPI1 | SPI3 | SPI6 | SPI12 | SPI1 | SPI3 | SPI6 | SPI12 | SSI1 | SSI3 | SSI6 | SSI12 | SSI1 | SSI3 | SSI6 | SSI12 |
| 1 | GEV | GEV | GEV | GEV | GEV | LP3 | GEV | GEV | LP3 | LP3 | GEV | LP3 | GEV | GEV | GEV | GEV |
| 2 | GEV | GEV | GEV | GEV | GEV | W2 | GEV | GEV | LP3 | LP3 | LP3 | GEV | GEV | GEV | LP3 | LP3 |
| 3 | GEV | GEV | GEV | LP3 | GEV | GEV | GEV | GEV | W2 | W2 | LP3 | GEV | GEV | GEV | GEV | GEV |
| 4 | LP3 | GEV | GEV | GEV | GEV | GEV | GEV | GEV | W2 | W2 | LP3 | GEV | GEV | GEV | GEV | GEV |
| 5 | LP3 | GEV | GEV | GEV | GEV | GEV | GEV | GEV | W2 | W2 | GEV | GEV | GEV | GEV | LP3 | GEV |
| 6 | | GEV | GEV | GEV | | GEV | LP3 | GEV | W2 | W2 | GEV | GEV | GEV | GEV | G2 | GEV |
| 7 | | GEV | GEV | GEV | | GEV | LP3 | GEV | W2 | W2 | GEV | GEV | GEV | GEV | LP3 | GEV |
| 8 | | | GEV | GEV | | GEV | GEV | GEV | GEV | GEV | GEV | GEV | | GEV | GEV | GEV |
| 9 | | | G2 | GEV | | GEV | GEV | GEV | LP3 | GEV | GEV | GEV | | GEV | GEV | GEV |
| 10 | | | GEV | GEV | | | GEV | GEV | GEV | GEV | GEV | GEV | | | GEV | GEV |
| 11 | | | GEV | GEV | | | GEV | GEV | GEV | GEV | GEV | GEV | | | GEV | GEV |
| 12 | | | | GEV | | | GEV | GEV | GEV | GEV | GEV | GEV | | | | GEV |
| 13 | | | | LP3 | | | | W2 | LN2 | GEV | GEV | GEV | | | | GEV |
| 14 | | | | LP3 | | | | | LN2 | GEV | GEV | GEV | | | | |
| 15 | | | | | | | | | LP3 | GEV | GEV | GEV | | | | |
| 16 | | | | | | | | | LP3 | GEV | GEV | GEV | | | | |
| 17 | | | | | | | | | | GEV | GEV | GEV | | | | |
| 18 | | | | | | | | | | GEV | GEV | GEV | | | | |
| 19 | | | | | | | | | | GEV | GEV | GEV | | | | |
| 20 | | | | | | | | | | GEV | GEV | GEV | | | | |
| 21 | | | | | | | | | | GEV | GEV | | | | | |
| 22 | | | | | | | | | | GEV | GEV | | | | | |
| 23 | | | | | | | | | | GEV | GEV | | | | | |
| 24 | | | | | | | | | | GEV | LP3 | | | | | |
| 25 | | | | | | | | | | | GEV | | | | | |
| 26 | | | | | | | | | | | GEV | | | | | |
| 27 | | | | | | | | | | | GEV | | | | | |

### 4.3 Deficit-IDF curves

Drought IDF curves of precipitation and streamflow deficits show general similarities in the two different climatic regions.
These curves are presented for each month separately at 1-, 3-, and 6-month time scales while the annual deficits are presented in one single set of IDF curves. For the sake of comparability of precipitation and streamflow deficits, various time scales and two different catchments, we presented deficits by dividing them by the respective precipitation threshold (PTH) and streamflow threshold (STH) (Figure 5).

The IDF curves show general similarities and, they are almost parallel to each other for return periods of 5 years and higher.
Deficits at the 2-year return period are separated from other return periods. Droughts have shorter durations at the 2-year return period. In all IDF curves, the precipitation and streamflow deficits decrease linearly as the drought duration increases. Yet, the



deficit IDF curves for the 2-year return period show a steeper decrease rate of streamflow deficit with duration in Kocher than in Seyhan and a relatively stronger decrease than the curves for higher return periods. Comparisons between the precipitation and streamflow deficit IDF curves show that in both catchments the drought durations of streamflow deficits are longer than those of precipitation deficits although precipitation deficits have more intense droughts compared to the thresholds. The thresholds change from one month to another and also the value of threshold changes notably from precipitation to streamflow (Figure 6). Short and more intense droughts are dominant in precipitation deficits while long and less intense drought are present in streamflow. In addition, precipitation deficits in summer are less than in winter considering the seasonality in the precipitation threshold with higher values in winter than summer. Precipitation deficits are lower during the summer months than the winter months in Seyhan because of the dominant Mediterranean climate's seasonality; nevertheless, no notable difference was observed between precipitation deficits in summer and winter in Kocher because of the lower seasonality in precipitation in the humid climate. With the deficit IDF curves, we found that the same drought intensity value calculated from the drought index corresponds to different precipitation and streamflow deficits at different stations and in different months of the year (Figure 6). This shows the spatial and seasonal variability of precipitation deficit or streamflow deficit. The general decrease with duration of deficit IDF curves for the 3-, 6- and 12-month time scales are similar to what we obtained at the 1-month time scale (Figures A1-A3).





**Figure 5.** Drought IDF curves in terms of precipitation and streamflow deficit divided by the threshold value for Seyhan and Kocher at 1-month time scale




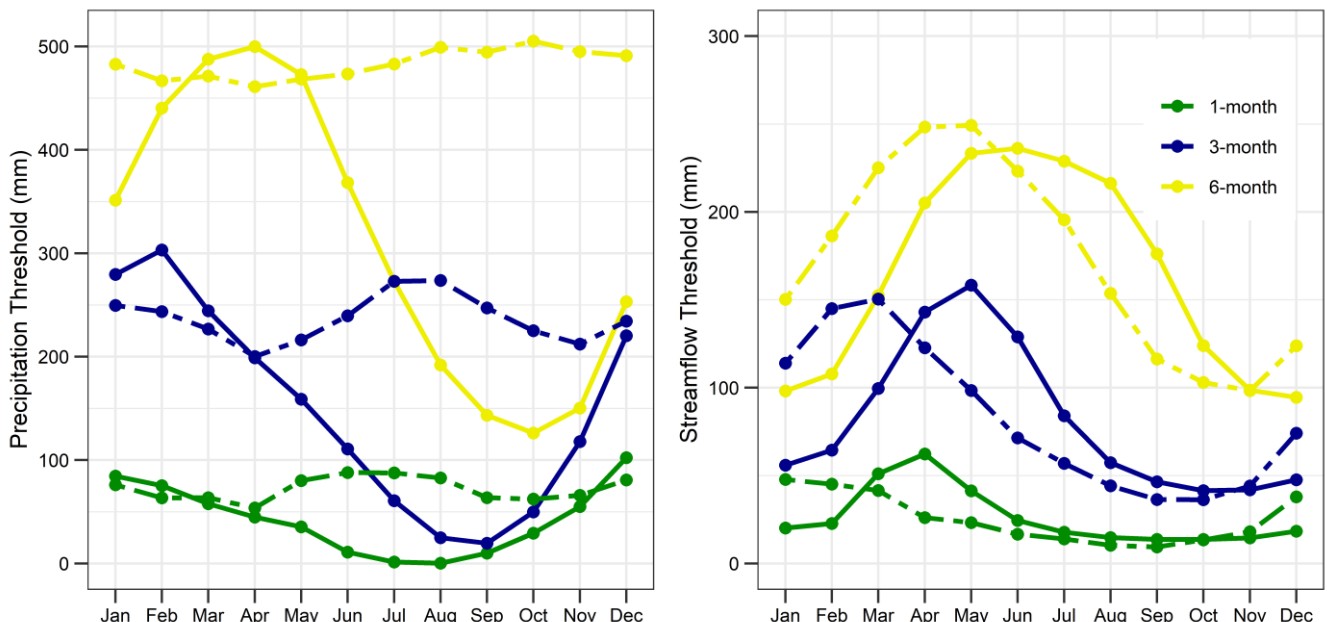

**Figure 6.** Precipitation threshold and streamflow threshold corresponding to $SPI_k = 0$ and $SSI_k = 0$. Solid lines are for Seyhan and dashed lines for Kocher

## 4.4 Observed major droughts in deficit IDF curves

The applicability of the drought IDF curves was tested for two specific drought events; the drought of 2008 in Seyhan River Basin and the drought of 2018 in Kocher catchment. By selecting the most critical drought of each duration, the observed drought events were plotted onto the drought IDF curves at the 12-month time scale (Figure 7). In the drought events, precipitation and streamflow deficits decrease as the drought duration increases. The precipitation deficit of the drought event in Seyhan corresponds to an event with about a 50-year return period at 1-month drought duration and around a 25-year return

period at longer durations. The return period declines with increase in the drought duration. However, streamflow deficit exceeds the 50-year return period in the most critical month. The drought in Kocher was a 25-year event at short durations and a 10-year event at longer durations in terms of precipitation deficit. Streamflow deficit IDF curves' relations of intensity, duration and frequency are generally similar to precipitation deficit curves. The drought event of 2018-2019 in Kocher streamflow had a return period higher than the 100-year for durations shorter than 4 months and higher than the 50-year for

longer durations. In both cases, the selected drought events approach the 100-year return period. In general, deficit in precipitation of a given return period is higher than streamflow deficit.





**Figure 7.** Drought IDF curves in terms of precipitation and streamflow deficits at 12-month time scale. The drought events of 2008 in Seyhan River Basin and 2018 in Kocher catchment were used to demonstrate the applicability of IDF curves



## 5 Discussion

### 5.1 Drought IDF curves for deficit indicators

We extended earlier studies on drought index (DI)-based IDF curves showing drought characteristics of a station in one single graph (Aksoy et al., 2021) to deficit-based monthly IDF curves. Based on those, this work first aimed to explore the question how these new curves relate to their non-dimensional index counterparts. Drought IDF curves based on precipitation and streamflow deficits for example provide the information how much precipitation and streamflow would be needed to mitigate negative impacts under the most critical drought condition. The IDF curves allow us to estimate return periods of a given deficit (or deficits of a given return period) using meteorological and hydrological drought indices. We showed how precipitation and streamflow deficits vary depending on the precipitation and streamflow characteristics of the region and the implications of this variation. IDF curves can provide similar information about the drought characteristics but physical variable IDF curves show that precipitation and streamflow deficits change in each month depending on the region's climate characteristics. Deficit values also change from one month to another at a station which means that one single IDF curve is not representative enough for the seasonal drought conditions. Drought IDF curves developed so far in literature were based on particular drought indices (Shiau and Modarres, 2009; Reddy and Ganguli 2012; Halwatura et al., 2015; Aksoy et al., 2021). Our findings and comparisons have critical implications for drought risk assessment. DI-based IDF curves are useful for the drought prediction at sites with no meteorological station or streamflow gauge. DI values such as SPI or its modified versions are reported by national meteorological services as a monthly routine in many countries. For example, the European Drought Observatory uses SPI to indicate a 'drought watch' situation (url1); The Turkish Meteorological Service (url2) and the German Weather Service (url3) use various drought indices to indicate spatial and temporal variation of drought at national level. At a more local scale, the low flow information service in Bavaria (url4) uses three different SPI indices (90-day, 30-day, 14-day) to monitor precipitation deficit and indicate drought situations. From the reported DI values of such monitoring maps, one may read the severity class of the drought in different return periods to prepare for the next months and plan or implement drought mitigation. An important limitation of DI-based IDF curves is that they mask the local seasonality. While there is no need to generate them for each month, the disadvantage is that the information they provide is not physically generic because the same value of DI is not equal to the same value of drought indicator, i.e., precipitation deficit or streamflow deficit for each month. Further steps are therefore needed to convert the drought index into deficits for use in practice. Deficit-based IDF curves are useful tools for drought research as well as for practical implementation. The user can read the return period of the observed drought from the IDF curve by inserting the observed precipitation and streamflow deficit on the curves, which allows practitioners particularly to act quickly and respond timely to mitigate the unwanted impacts of drought.

### 5.2 IDF curves for drought mitigation planning in different climatic

The work further aimed to compare how precipitation and streamflow deficits of drought indices and their associated drought event characteristics differ both between the variables within a basin and between different climatic regions. The two events





shed some light on these differences. Tijdeman et al. (2022), for example, described the drought event of 2018 in southern Germany as multi-year deficit event that a short summer extreme was superimposed on. The intensities determined here confirm this notion with the event's shorter duration deficits corresponding to higher return periods on the IDF curves, i.e.,

more extreme conditions than the longer durations. The differences in intensity over duration are mirrored in streamflow, but less pronounced. Overall, the meteorological conditions appear to have combined into an event that corresponds to a substantially more extreme (less frequent) streamflow deficit than those of its meteorological causes. The curves therefore might help explain different drought types as well. The drought event of 2008 at Seyhan while also corresponding to higher return periods for streamflow than for precipitation, varies differently regarding the return periods over different durations.

With increasing duration, the corresponding return period of this drought decreases for precipitation but increases for streamflow. Possibly, this might be due to either the increased water use having amplified the streamflow deficit or higher evapotranspiration and the missing seasonal precipitation.

DI-based drought IDF curves do not incorporate site specific precipitation and streamflow deficits of each month other than the drought indices. However, regional precipitation patterns, including intense and prolonged deficits, play a critical role in

identifying frequency of drought and challenge to mitigate drought by established management plans (Kreibich et al., 2022). Short and long-term deficits can be detected from the intensity of the drought indicators combined with duration and frequency. In particular, long-term droughts with high variations and frequently occurring intense deficits can be assessed as primary characteristics for determining site sensitivity, while regular and comparatively short deficits are general characteristics of the regional climate condition (Halwatura et al., 2015). The different patterns of summer precipitation and streamflow in the two

case studies, Seyhan and Kocher, illustrated how seasonality become the primary factor of deficits. We found that short time scale deficits are the most severe droughts with higher variability. The lower variability of longer time scales applied both to precipitation and streamflow. Implementation of the IDF curves on the drought events of 2008 in Seyhan and 2018 in Kocher demonstrated this reality and the impact of drought in these regions (Cavus et al., 2022; Tijdeman et al., 2022; Rakovec et al., 2022).

**5.3 IDF curves for practice: time scale and severity**

Deficit-IDF curves for the design under drought conditions have not been well established in literature and practice. There is an emerging need for guidelines worldwide to use in water resources design, planning, operation, and management under low flow or drought conditions (Vogel and Kroll, 2021). Similar to the precipitation IDF curves used in hydrological practice (Chow et al., 1988), deficit-IDF curves developed in this study can therefore be considered an improvement towards the design

under drought conditions. In practical meaning, the time scale is the time lag between the initiation of water deficit and its impact on water resources, engineering activity, ecology, economy or society. An assessment about the choice of the time scales has been made by Vicente-Serrano et al. (2013) for vegetation, and Halwatura et al. (2015) for ecosystem establishment after mining. They divided the time scales into two; short and long time scales and provided a list of dominant time scales. However, the time scales can overlap and change from region to region or from season to season in the same region. For





example, a short but severe drought might have the same impact as a long mild drought. For these reasons, we consider the applicability of IDF curves by time scale and severity of various activities.

Successful drought and water management require knowledge of drought characteristics at varying time scales. Seasonal or even over-year water storage and release at longer time scales are important processes in some catchments and for some water uses while in other catchments for other water needs shorter time scale variability matters as water needs to be provided

regularly, e.g., for hydropower generation. Respectively, different drought threats matter as the impacts are very different and this will affect the usefulness of the IDF curves. This final section explores how IDF curves might be beneficial for management actions such as agriculture-irrigation for different crops, industrial water uses, and reservoir-storage for different purposes or run-off-the-river hydropower, as well as for nature's water needs, such as aquatic ecology. Table 3 categorizes these water needs and the associated drought time scales and drought severity classes that threaten them.

Irrigation methods for agricultural activities require cost-effective drought management strategies for specific sites. The IDF curves of precipitation deficit of long time scales and moderate/severe droughts can provide a useful knowledge for larger area agriculture such as the design of drainage systems and irrigational canals (Table 3). For drip, seasonal and critical stage irrigation for which a lower amount of water is needed compared to larger drainage systems, short time scales provide important information about the deficit. Horticulture and field crops cannot resist against long-term deficits; i.e., any design

for such activities should consider short time scales. For non-irrigated agriculture, precipitation deficit IDF can be used at short and long time scales while for irrigated agriculture, streamflow deficit IDF curves can be used at different time scales depending on the region and crop type. Long time scale IDF curves can be considered for drought-tolerant species with deep roots. Regarding crop species, some specific crop types cannot grow if the drought exceeds a certain duration or intensity. For instance, precipitation deficit-IDF curves at short time scales are important for drought intolerant crops while long time scales

are important drought tolerant crops, annual and perennial grasses, trees.

**Table 3.** Time scale and severity classes of droughts for some specific activities

| General category | Specific activity | Drought time scale and severity |
|---|---|---|
| Agriculture - Irrigation | Drainage system, irrigation canals, drip irrigation | SM, SS, LM, LS |
| | Seasonal and critical stage irrigation | |
| Agriculture-Rainfed crop species | Drought intolerant crops, horticulture, field crops | SM, SS |
| | Drought tolerant crops, annual and perennial grasses, trees | LS |
| Hydropower | Small (runoff) hydropower facilities, thermal power plants | SM, SS |
| | Large (reservoir) hydropower facilities | LM, LS |
| Reservoir storage | Water supply (depending on the reservoir size and water demand) | SS, LM, LS |
| | Irrigation (depending on the season and crop) | SS, LM, LS |
| Industry | Industrial water | SS, LM, LS |
| Ecology | Recreation, ecosystem habitats | SS |



*SM - Short time scale and mild/moderate drought; SS - Short time scale and severe/extreme drought; LM - Long time scale and mild/moderate drought; LS - Long time scale and severe/extreme drought * S: Shorter than 6 months, L: 6 months or
longer

Diversion hydropower plants or thermal power plants as they often exist in Germany and also further downstream in the Kocher will be affected by droughts at short time scales while storage hydropower plants as they often exist in the mountains or in semi-arid climates in the Mediterranean region, will be interested in the threat from long droughts (Table 3). Short time
scale IDF curves based on streamflow deficit can provide useful information under mild/moderate drought conditions for river water uses for industry or for recreational use. Ecosystem habitats again might mostly require consideration at short time scales but during very specific seasons (spawning, migration time, oxygen depletion), but also longer periods when it comes to overall environmental degradation. The return periods of streamflow deficits provide the probability of deficits at these durations and intensities, and the risk of drought can be interpreted correctly. This is important for water managers who can conduct a cost-
benefit analysis whether the cost of taking some mitigation precautions such as agriculture, reservoir, hydropower etc. are comparable with the cost of probable failure (Table 3). Selection of small hydropower management based on different time scales is also the key management actions for deficit-IDF curves.

Another issue could be related to the variability. A less variable system can be linked with longer time scale, and a more variable system with a shorter time scale. Specifically, short time scale streamflow deficit IDF curves should be used for the
run-of-the-river and pumped storage energy production plants, because they have higher fluctuation in the river. The storage systems can be tolerant for long time scale droughts if the storage is filled enough at the beginning of dry period. Therefore, long time scale streamflow deficit IDF curves work well for storage hydropower systems. For water supply systems, streamflow deficit IDF curves at short time scales can be advised. The categorization outlined in this study (Table 3) clarifies the usefulness of the deficit-based IDF curves to address the site-specific climatic conditions. Drought IDF curves should be a
primary factor of drought mitigation strategies and eventually help to guide water recourses management planning where hydrological extremes have an impact on current operations.

## 6 Conclusion

We developed deficit-IDF curves to assess how associated deficits vary in a given catchment at different return periods although they have similar drought intensity in the drought index. We explained the advantage of using physical variables,
precipitation and streamflow deficits, giving more practical and straightforward use to the deficit IDF curves than the index-based drought IDF curves. This makes the new drought IDF curves generally applicable for different climatological regions and also for a comparison of their drought risks. Deficit IDF curves can be developed for any location or catchment to convert drought index values to deficit. Precipitation and streamflow deficits distinguish the degree of drought impact and its implication for various water uses. The use of IDF curves with the observed droughts showed their applicability in identifying





return periods. A given value of both deficit has higher return period at longer drought duration and lower return periods at shorter drought durations. Precipitation deficits of the observed major droughts are higher in amount but have lower return periods than their streamflow deficits. Here, we stress that deficits can be highly variable, which make it necessary to comprehensively assess the usefulness of IDF curves before establishing drought mitigation strategies. Thanks to the deficit knowledge, drought intensity in a particular system can be interpreted correctly, especially those characterized by complex hydrological systems. Deficit IDF curves can be applied to quantify the frequency of drought events and characterized by intensity and duration at different time scales. Precipitation and streamflow deficits of droughts should be used to assess drought management strategies based on return periods.

*Author contribution*. Yonca Cavus: Data curation, Formal analysis, Investigation, Methodology, Software, Validation, Visualization, Writing - original draft, Writing - review and editing. Kerstin Stahl: Investigation, Methodology, Supervision, Writing - review and editing. Hafzullah Aksoy: Investigation, Methodology, Supervision, Writing - review and editing.

*Acknowledgement.* The study is supported by the DAAD "Research Grants - Bi-nationally Supervised Doctoral Degrees / Cotutelle" Program for which the authors are thankful. We thank all data providing organizations named in the Data Availability Statement.

*Competing interests*. The contact author has declared that neither they nor their co-authors have any competing interests.

*Data Availability Statement.* Precipitation data provided by the Turkish State Meteorological Service (TSMS) can be purchased from https://mevbis.mgm.gov.tr/mevbis/ui/index.html. Precipitation data provided by the German Weather Service (DWD) can be downloaded from this link (https://opendata.dwd.de/climate_environment/CDC/observations_germany/). Streamflow data provided by the general directorate of State Hydraulic Works (DSI) can be downloaded from this link (https://www.dsi.gov.tr/Sayfa/Detay/744#). Streamflow data provided by the Environment Agency of Baden-Württemberg (LUBW) can be obtained on request (https://www.lubw.baden-wuerttemberg.de/startseite).






**Figure A1.** Drought-IDF curves in terms of precipitation and streamflow deficit divided by the threshold value for Seyhan and Kocher at 3-month time scale







**Figure A2.** Drought-IDF curves in terms of precipitation and streamflow deficit divided by the threshold value for Seyhan
and Kocher at 6-month time scale



**Figure A3.** Drought-IDF curves in terms of precipitation and streamflow deficit divided by the threshold value for Seyhan and Kocher at 12-month time scale

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
