# Peer review of "Drought intensity-duration-frequency curves based on deficit in precipitation and streamflow for water resources management"

_Hydrology and Earth System Sciences, 2023_

## Author Response (AR2)

**Executive editor decision:** Publish subject to technical corrections
by Giuliano Di Baldassarre

**Comments to the author:** The paper can be published after addressing the editor's comments.

**Editor decision:** Publish subject to technical corrections
by Khalid Hassaballah

Dear Authors,

Thank you for submitting the revised manuscript. The comments raised by the two referees are well addressed. However, there are still few technical corrections to be made:

Technical correction:

Line 87: the revised text is not same as suggested revision in the text. Please remove "we used" to minimize the use of many 'we' as requested by referee 1.

In step 1, as a response to the referee 1 related to probability distribution functions, you have mentioned in your suggested revised text "Pearson Type 3" as one of the tested distributions. However, this was not mentioned in the revised manuscript and was not shown in the new added figure3!. If the Pearson Type 3 was not among the tested distributions, please check the following step 2 to confirm what is written there is correct.

Sincerely Yours
Khalid Hassaballah

We are thankful to Editor and Executive Editor for their positive decision on our paper. We responded two technical corrections raised by the Editor as follows:

1. Line 87: We removed the word "we used" from the text as suggested.
2. We are sorry about this confusion. You are right that we have listed "Pearson Type 3" among the tested distributions in the response to the referee 1. This has been done unintentionally and therefore we removed it from the list. We confirm that the "Pearson Type 3" was not among the tested distributions and the revision we have made is correct. Please note that we have provided a better quality image version of Figure 3.

Sincerely Yours
Yonca Cavus

**---------------------- RESPONSES TO THE COMMENTS ----------------------**

We appreciate the constructive and prompt additional comment of the Editor. Our response to this comment is given below. The comment is in **black** while our response is given in **blue** and any citation how we revised the text in the revised manuscript in **red**.

*Please note that this document covers our responses to the comments of the Editor and anonymous reviewers and revisions made in the manuscript accordingly.*

**Editor:**

The paper received two independent referees; it became clear that the paper needs minor revisions before it can be accepted for publication.
I appreciate the point-by-point response. The points raised by the referees have been suitably addressed in your replies. Please change your manuscript accordingly.
In addition, please:
- comment of Reviewer1#: Lines 135-137: It is known that Gamma distribution is used for SPI but it is advised to make a clear reference to McKee et al. (1993). How did you find that the General Extreme Value (GEV) is the best-fit probability distribution function for the standardized streamflow index (SSI)? Among which distributions?
In addition to the proposed revision in the text, I suggest you also add the plotting that present the Anderson-Darling statistical test results for different probability distribution functions to show that the Generalized Extreme Value (GEV) has the best-fit probability distribution function.

**Response:** We are glad that our work on the anonymous reviewers' comments is appreciated. Also, we are very thankful for the additional constructive comment which allowed us to clarify our choice of GEV as a best compromise if one wants to use the same distribution for all (rather than absolute best-fit in absolute terms as may have been mistaken in the current draft). We kept the revision short as we do not think that the details should be in the main document in order not to shift the focus of the paper towards this technical aspect of distribution fitting and comparison.

The comment also allowed us to correct the sentence (in the original text) indicating the Generalized Extreme Value (GEV) as the best-fit probability distribution function of the streamflow volume accumulated over a month for the two catchments. Among the tested probability distribution functions, GEV and Lognormal (LN2) came to the front for both catchments. GEV is the best and LN2 the second for Seyhan, LN2 is the best and GEV the second for Kocher for the streamflow volume accumulated over a month. GEV was chosen for both catchments for comparability reasons and also to be in accordance with the literature (Vicente-Serrano et al., 2012). This choice has already been stated in the original paper and we hope we clarified it more in the revision.

As suggested, we plotted probability distribution functions based on the Anderson-Darling statistical test and provided as a figure (Figure 3). It is referenced in our response to the comment. This is also linked to the comment of Reviewer #1 which is now addressed and revised as follows (Note that Figures in the manuscript were re-numbered accordingly):

**Revision:** The Gamma probability distribution function is fitted to the precipitation data and transformed to the standard normal distribution with zero mean and unit standard deviation (McKee et al., 1993). Based on the Anderson-Darling statistical test of several probability distribution functions including Gamma, Generalized Logistic, GEV, Gumbel, Logistic, Lognormal (LN2), Normal, and Weibull, we found the Generalized Extreme Value Distribution (GEV) the best, LN2 the second for Seyhan, and LN2 the best, GEV the second for Kocher for the streamflow volume accumulated over a month (Figure 3). The length of the streamflow

record and the selected probability distribution function strongly affect the calculated SSI (Wu et al., 2005, Tijdeman et al., 2020). To allow a comparison of the streamflow deficits in the two different case study rivers in different climate regions, here we used GEV for both catchments for comparability reasons and also to be in accordance with the literature (Vicente-Serrano et al., 2012).

[Figure]

Figure 3. Probability distribution functions fitted to streamflow volume for selected gauging stations in Seyhan River Basin and Kocher catchment.

**Reviewer #1:**

**General Comments**

Drought has been widely evaluated by using different standardized drought indices in the literature. Some studies take the drought indices to develop drought intensity-duration-frequency (IDF) curves as reviewed by the authors in this paper. It is emphasized that the drought indices are dimensionless, and it is therefore difficult to interpret them physically, meaning that precipitation and streamflow deficits cannot be quantified directly from the drought index-based drought IDF curves. This is a true argument. At this point, this paper aims to develop drought IDF curves in terms of precipitation and streamflow deficits to overcome this difficulty. The proposed methodology was applied on two different climatic regions, one from Turkey and another from Germany. The applicability of the IDF curves has been shown by using historical droughts experienced in both regions. Another remarkable result in this study is Table 3 which represents the practical value of IDF curves for various sectors. In my opinion, the deficit-IDF curves developed in this study will be useful tools in establishing proper drought mitigation strategies. Therefore, I recommend publication of the paper after minor revision based on the following specific comments and technical corrections.

**Response:** We are thankful for the clear and constructive comments. We took them all into account in detail and responded each as listed below:

**Specific Comments (in the order they appear in the text)**

- Lines 12-13: Revise to clarify the statement "Drought characteristics assigned the same value in index are not necessarily the same in different regions, and in different months of the same region" as it is not clear.

**Response:** We revised this statement as follows:

**Revision:** Drought severity calculated from the drought index does not necessarily correspond to the same amount of deficit in precipitation or streamflow at different regions and it is different for each month in the same region.

- Line 96: There is no need to emphasize the ratio of the agricultural lands (51%) in Kocher. The paper in not agriculture-oriented and results are not affected by this particular ratio. The previous sentence good enough to state that the catchment has importance for agriculture as it was stated for Seyhan previously in this paragraph.

**Response:** We removed this sentence from the text in our revised document.

- Lines 135-137: It is known that Gamma distribution is used for SPI but it is advised to make a clear reference to McKee et al. (1993). How did you find that the General Extreme Value (GEV) is the best-fit probability distribution function for the standardized streamflow index (SSI)? Among which distributions?

**Response:** We revised the statement on the Gamma distribution in the text by referencing to McKee et al. (1993). Regarding calculation of SSI, we used the SCI package in R which uses several probability distributions such as Gamma, Generalized Logistic, GEV, Gumbel, Logistic, Lognormal, Normal, and Weibull. We tested all of these distributions on the streamflow data and found the GEV the best-fit probability distribution function after the Anderson-Darling statistical test. This was also commented by the Editor. We updated the revision as follows:

**Revision:** The Gamma probability distribution function is fitted to the precipitation data and transformed to the standard normal distribution with zero mean and unit standard deviation

(McKee et al., 1993). Based on the Anderson-Darling statistical test of several probability distribution functions including Gamma, Generalized Logistic, GEV, Gumbel, Logistic, Lognormal (LN2), Normal, and Weibull, we found Generalized Extreme Value (GEV) the best, LN2 the second for Seyhan, and LN2 the best, GEV the second for Kocher for the streamflow volume accumulated over a month (Figure 3). The length of the streamflow record and the selected probability distribution function strongly affect the calculated SSI (Wu et al., 2005, Tijdeman et al., 2020). To allow a comparison of the streamflow deficits in the two different case study rivers in different climate regions, here we used GEV for both catchments for comparability reasons and also to be in accordance with the literature (Vicente-Serrano et al., 2012).

- Line 170: Can you give the name of any other curve tested for establishing functional relationship between the drought indicators and indices?

**Response:** We tested a set of functions among which we chose the logistic curve because it fitted better than others and it was capable to deal with zero precipitation. We revised the text accordingly as follows.

**Revision:** To establish a functional relationship between the drought indicators and the drought indices, we tested several curves including the second and third order polynomials, exponential, Gompertz and logistic curve. The polynomials, exponential and Gompertz curves were discarded as they produced negative values of precipitation or streamflow, and they could not fit properly to SPI or SSI time series. The logistic curve was selected as the best-fit curve because it provided the highest correlation without producing negative precipitation and streamflow values (Sit and Poulin-Costello, 1994). The logistic function is given by

- Line 183: It is not clear what $DI_d$ in Eq. 2 is. Please make the statement clear.

**Response:** $DI_d$ represents deficit in the drought indicator. We revised the text by also adding "$DI_d$" in parenthesis to clarify.

**Revision:** The difference between the threshold drought indicator and the critical drought severity is deficit in the drought indicator ($DI_d$), which is either precipitation deficit or streamflow deficit. For each duration and return period, it is calculated by

- Figure 6: The 12-month threshold, which is a constant value over the year is not given in Figure 6. It can be added to Figure 6 as a line or be indicated in the figure caption.

**Response:** We preferred to revise the figure caption as suggested in order not to lose details about the seasonality because of higher values of thresholds at the 12-month timescale.

**Revision:** Figure 6: Precipitation threshold (PTH) and streamflow threshold (STH) corresponding to $SPI_k = 0$ and $SSI_k = 0$ ($k = 1, 3, 6$ months). Solid lines are for Seyhan and dashed lines for Kocher. The 12-month thresholds are constant values over the year; PTH = 643.88 mm, STH = 333.39 mm for Seyhan; and PTH = 975.57 mm, STH = 356.18 mm for Kocher.

- Lines 243-266: Precipitation threshold and streamflow threshold (in Figure 6) are used in the development of IDF curves (in Figure 5). Thus, I suggest that Figure 6 comes before Figure 5. The text about Figures 5 and 6 needs a thorough check. Please revise your text by considering that Figure 6 tells the story about the within year variability of precipitation and streamflow thresholds and Figure 5 is a tool based on this story. Also, probably because of the disorganization of Figures 5 and 6, I found some statements unclear or redundant: (a) Omit statement 'Comparisons between the …' starting in Line 253. It is a sentence hard to understand and to me again the statement about the drought duration is

not true. The second part coming after 'although' is confusing. (b) Revise statement 'Short and more …' starting in Line 257. While revising pay attention to replace 'drought in precipitation deficit' with 'drought in precipitation'. (c) Delete statement 'With the deficit …' starting in Line 262. It reflects an opinion about the IDF curves rather than being result. This is well fit to the Discussion Section (subsection 5.1) where this fact has already been emphasized in Lines 312-314. There is no need to have it here.

**Response:** We agree that we should reorganize this part of the paper by removing redundant sentences.

**Revision:**
4.3 Deficit IDF curves
Precipitation and streamflow thresholds are needed to calculate deficits in precipitation and streamflow, the key elements of the deficit-IDF curves. The thresholds are not constant over the year at 1-, 3- and 6-month timescales (Figure 5). They change from one month to another and also the value of the thresholds changes notably from precipitation to streamflow. The annual thresholds are constant values as they accumulate the within-year seasonality. The clear seasonal variability of precipitation threshold in Seyhan is not evident in Kocher while the streamflow threshold has a notable seasonality in both catchments. These temporal and spatial variabilities in the thresholds prevent the drought IDF curves from being comparable in time and space. For the sake of comparability of the IDF curves of various timescales at two different catchments, precipitation deficit and streamflow deficit were divided by the respective precipitation threshold (PTH) and streamflow threshold (STH), respectively.

Drought IDF curves of precipitation and streamflow deficits are presented for each month separately at 1-, 3-, and 6-month timescales (Figure 6, Figures A1-A2) while the annual deficits are presented in one single set of IDF curves (Figures A3). Drought IDF curves of precipitation and streamflow deficits show general similarities in the two different climatic regions. In all IDF curves, the precipitation and streamflow deficits decrease linearly as the drought duration increases. The IDF curves are almost parallel to each other for return periods of 5 years and higher. Deficits at the 2-year return period decrease faster, they are therefore separated from other return periods. However, the 2-year return period shows a steeper decrease rate of streamflow deficit in Kocher than in Seyhan and a relatively steeper decrease than the curves for higher return periods. Compared to the precipitation and streamflow thresholds, droughts in precipitation are more intense than in streamflow. In addition, precipitation deficits in summer are less in absolute values than in winter considering the seasonality in the precipitation threshold with higher values in winter than summer. Precipitation deficits are particularly lower during the summer months than the winter months in Seyhan because of the dominant seasonality in the Mediterranean climate while no notable difference was observed between precipitation deficits in summer and winter in Kocher because of the negligible seasonality in precipitation in the humid climate of this catchment. The linear decrease in deficits with increasing drought duration and the faster decay in the 2-year return period for the 3-, 6- and 12-month timescales (Figures A1-A3) are similar to what we obtained at the 1-month timescale (Figure 6).

**Technical Corrections**

▪ Lines 80-85: So many "we" in this paragraph.

**Response:** We agree to minimize the use of 'we' and to revise the paragraph.

**Revision:** The overarching objective is to develop drought IDF curves based on precipitation and streamflow deficits at different timescales and appraise their usefulness in different climates. For this, drought index-derived characteristics are converted into precipitation and streamflow deficits for different cases. For comparison, two catchments are considered in

different climatic regions. Hence, the comparison between the regions and among the variables provides the framework of analysis. Specifically, we aim to assess how the deficits vary in the considered study regions and finally explore how deficit IDF curves might be used for different purposes. The study addresses the following research questions:

- Line 116 (Caption of Table 1): Put the name of countries in parenthesis as (Turkey) and (Germany) after each catchment to make the table caption self-explanatory.

**Response:** We agree to add the name of countries into the caption of Table 1.

**Revision:** Table 1. Characteristics of selected meteorological and streamflow gauging stations in Seyhan (Turkey) and Kocher (Germany)

- Line 118 (Figure 1): For a better visualization, name of the regions (Seyhan and Kocher) should be centered at the top their group of graphs instead of giving them in the precipitation graph only.

**Response:** We agree to replot Figure 1 by centering name of regions in the graphs.

[Figure]

- Line 177: DIc should be italic.

**Response:** We agree to correct $DI_c$ as Italic. Thank you so much for being so precise.

- Line 225 (Caption of Figure 4): Please add the following sentence to the caption to clarify horizontal axis: 'Horizontal axis indicate SPI and SSI values at the top of each scatter diagram.'

**Response:** We agree to add this sentence to the caption of Figure 4. Thank you for this clarification.

**Revision:** Figure 4. The relation between precipitation as drought indicator and SPI, and streamflow as drought indicator and SSI at 1-, 3-, 6- and 12-month timescales for Seyhan and Kocher. Horizontal axis indicates SPI and SSI values at the top of each scatter diagram.
- Line 257 (end): Make plural as "… intense droughts …"

**Response:** We agree to correct it in the text. Thank you for being so precise.

- Lines 378-380: The footnote is related to Table 3 but there is no "*" in the table. The authors may consider putting it into the Table caption instead of keeping it as a footnote to avoid such confusion.

**Response:** We suggest to move the footnote to the caption of Table 3.

**Revision:** Table 3. Timescale and severity classes of droughts for some specific activities. SM - Short timescale and mild/moderate drought; SS - Short timescale and severe/extreme drought; LM - Long timescale and mild/moderate drought; LS - Long timescale and severe/extreme drought; S: Shorter than 6 months, L: 6 months or longer

**Reviewer #2:**

**General Comments**

The manuscript presents a methodology for development of intensity-duration-frequency (IDF) curves based on the frequency analysis of severity of the critical (the most severe) droughts of different fixed durations based on two non-dimensional indices, SPI and SSI, and their consequent transformation into corresponding dimensional quantities, deficits of precipitation and streamflow. In this way, final IDF curves are easier to interpret in various contexts in which drought analysis is needed.

The overall presentation of the methodology in the paper is clear, with a minor shortcoming that I describe as the first specific comment. The motivation for proposing this methodology is laid out well. The part of the discussion related to the potential for the use of the IDF curves provides an excellent overview. In my opinion, the discussion and conclusion sections in the paper lacks two aspects that I describe as the second and the third specific comment.

I find that the proposed approach, which considers different fixed drought durations, provides a more rigorous structural description of droughts than other approaches where duration is treated as an additional variable. I therefore commend the authors on devising this methodology.

Overall, I propose minor revision of the manuscript in accordance with the comments below.

**Response:** We are thankful for the clear and constructive comments. We took them all into account in detail and respond to each below:

**Specific comments**

1. Description of Step 4 of the methodology (around L170) lacks specification of data used in developing logistic regression between SPI/SSI and precipitation/streamflow. Is this based only on the critical droughts or on all droughts? What was the time scale of precipitation/streamflow used in the regression analysis (matching the time scale of SPI/SSI or something more general, like annual precipitation)? I suggest that this is precisely described so that the methodology can be fully comprehended and potentially replicated for other sites by other authors.

**Response:** We agree to revise this sentence to make it clear. Each month in the time series has a drought indicator (precipitation or streamflow) accumulated at 1-, 3-, 6-, and 12-month timescales and the corresponding drought index (SPI or SSI) at the same timescales.

Regression equations were developed between the drought indicators and the indices for each timescale separately by using all indicators and indices. Due to the periodicity, the regression equations were fitted separately for each month of the year; 12 regression equations were obtained for 1-, 3-, 6-month timescales, and one regression equation for 12-month timescale. We inserted this information as in the statement below to come after Equation (1) at the end of paragraph explaining Step 4 of the methodology (around Line 170).
**Revision:**

$$f(x) = \frac{a}{1 + be^{-cx}}$$ (1)

in which $x$ is the drought index, and $a$, $b$ and $c$ are parameters. Each month in the time series has a drought indicator (precipitation or streamflow) accumulated at 1-, 3-, 6-, and 12-month timescales and the corresponding drought index (SPI or SSI) at the same timescales. Regression equations were developed between the drought indicators and indices by using their all values. Owing to periodicity (seasonality), the relation changes from month to month for timescales shorter than a year; 12 regression equations were obtained for 1-, 3-, 6-month timescales, and one regression equation was obtained for the 12-month timescale.

2. The methodology builds on droughts defined using the thresholds SPI = 0 and SSI = 0. However, specific areas for which the droughts need to be analysed may require different types of thresholds. For example, in agriculture, different crops have different water requirements in terms of precipitation. How would your methodology fit to that type of thresholds? Furthermore, can the proposed methodology be easily adapted to other indices than SPI and SSI?

**Response:** The methodology is flexible to use with different threshold and it can be adopted to other indices. We added following statement to discuss the threshold issue and the extension of the methodology to other drought indices as commented.

**Revision:** Deficit IDF curves can be applied to quantify the frequency of drought events and characterize the droughts by their intensity and duration at different timescales. The threshold level is also important to consider in the IDF curves which is taken as the level corresponding to SPI = 0 and SSI = 0 in this study. By doing so, we considered to take all drought classes (extreme, severe, moderate, and mild drought) of McKee et al. (1993) in the methodology which is flexible to use with any other threshold. The methodology can also be adapted to other drought indices than SPI and SSI. It is possible to choose a lower threshold level to exclude mild droughts for which a new set of IDF curves will be obtained as the IDF curves are threshold-dependent.

3. In my opinion, the conclusions lack a brief overview of open questions and potential for further research. These open questions could be regional IDF curves, estimation at ungauged sites etc.

**Response:** We agree to extend final sentences in the conclusion section by adding open questions and potential future research.

**Revision:** Deficit IDF curves can be applied to quantify the frequency of drought events and characterize droughts by their intensity and duration at different timescales. The application here was limited to a few examples regarding indices, timescales, and thresholds for station-based precipitation and streamflow deficits of droughts. The deficit IDF curve approach of this study can also be adapted to drought indices other than SPI or SSI, and to threshold levels corresponding to different situations in terms of severity or even to impact-specific thresholds. How transferable the approach is to other indices and thresholds and also to other climates and hydrological regimes than the examples used here remains to be tested in order to assess

the range of applicability. Further work might also explore the extension of station-based drought IDF curves to develop regional curves for a possible use at ungauged basins.

**Minor comments and technical corrections**

L72-74: Sentence starting with "Because…" is unclear or unfinished. Please revise.

**Response:** We found that this sentence interrupted smooth reading of the paragraph. We moved this sentence to the beginning of its paragraph in Line 67 (after omitting the word 'Because') to read as follows:

**Revision:** Meteorological and hydrological droughts correspond to temporal anomalies changing also spatially from one catchment to another and they are characterized based on long-term conditions, which are related to climatic and environmental factors (Vicente-Serrano et al., 2013; van Loon, 2015). Therefore, the same value of a drought index corresponds to different deficits in different regions. The non-dimensionality of drought indices comes at the expense of physical non-interpretability, i.e. most drought indices cannot be read quantitatively as actual precipitation and streamflow deficits. In climates with high seasonal variation (i.e., Mediterranean climate), the difference between deficits varies greatly in each month while this difference may be lower in regions with low seasonality (i.e., humid climate). Determination of precipitation and streamflow deficits is a challenge when the common drought indices are used. Thus, any non-dimensional drought severity or intensity derived from index series might insufficiently represent the actual water availability for water management under drought conditions.

L113: Replace "likely to observe" with "likely to be observed".

**Response:** We agree to correct it in the text. Thank you for being so precise.

L153: Remove "for each year" from this sentence. Frequency analysis is done for the series of annual maximum severities, but not for each year.

**Response:** We agree to remove it in the text. Thank you for this correction.

L175: Please check this part: "drought INDICATORS were converted to precipitation and streamflow deficits using the relation between the drought indicators and indices". Looks like the first "indicators" should be replaced by "indices".

**Response:** Yes, it is a mistyped word. We will change 'indicators' with 'indices' in the text. Thank you so much for the careful correction.

L319: Section 5.2 heading is incomplete: "… in different climatic" what? (maybe conditions?)

**Response:** We agree to complete the heading as 'different climatic conditions' in the text.

L393: I suggest replacing "variability" with "temporal variability" (since one could also possibly discuss spatial variability of droughts).

**Response:** We agree to specify the variability by adding 'temporal'. Thank you for being so precise.